# A Novel Line-Scan Algorithm for Unsynchronised Dynamic Measurements

Simon Verspeek [1,†] , Thomas De Kerf [1,*,†] , Bart Ribbens [1] , Xavier Maldague [2] , Steve Vanlanduit [1] and Gunther Steenackers [1]

1   Faculty of Applied Engineering, Department Electromechanics, Research Group InViLab, University of Antwerp, Groenenborgerlaan 171, B-2020 Antwerp, Belgium; simon.verspeek@uantwerpen.be (S.V.); bart.ribbens@uantwerpen.be (B.R.); steve.vanlanduit@uantwerpen.be (S.V.); gunther.steenackers@uantwerpen.be (G.S.)
2   Computer Vision and Systems Laboratory, Department of Electrical and Computer Engineering, Université Laval, Quebec City, QC G1V 0A6, Canada; xavier.maldague@gel.ulaval.ca
*   Correspondence: thomas.dekerf@uantwerpen.be
†   These authors contributed equally to this work.

**Abstract:** In non-destructive inspections today, the size of the sample being examined is often limited to fit within the field of view of the camera being used. When examining larger specimens, multiple image sequences need to be stitched together into one image. Due to uneven illumination, the combined image may have artificial defects. This manuscript provides a solution for performing line-scan measurements from a sample and combining the images to avoid these artificial defects. The proposed algorithm calculates the pixel shift, either through checkerboard detection or by field of view (FOV) calculation, for each image to create the stitched image. This working principle eliminates the need for synchronisation between the motion speed of the object and the frame rate of the camera. The algorithm is tested with several cameras that operate in different wavelengths (ultraviolet (UV), visible near infrared (Vis-NIR) and long-wave infrared (LWIR)), each with the corresponding light sources. Results show that the algorithm is able to achieve subpixel stitching accuracy. The side effects of heterogeneous illumination can be solved using the proposed method.

**Keywords:** non-destructive inspections; dynamic line-scan thermography; ultraviolet measurements; dynamic measurements; multispectral imaging

## 1. Introduction

Non-destructive testing (NDT) is an attractive way to evaluate the performance, integrity, or deterioration of an object or structure. With non-destructive testing, the object or structure being inspected does not have to be disassembled to find hidden features that are not visible on the surface. Another advantage is that, compared to destructive analysis methods such as fracture or accelerated corrosion testing, the inspection can be performed without affecting the material itself.

There are several methods of non-destructive testing, such as magnetic particle inspection (MPI) [1], radiographic testing, ultrasonic testing (UT) and eddy-current testing (ECT). However, there are several drawbacks using these methods, namely, MPI can only inspect metallic samples, RT requires extensive safety precautions and UT and ECT are point-based methods which are not suitable for a large sample size. To compensate the aforementioned drawbacks, camera-based methods can be used in a wide variety of applications. Camera-based methods include infrared thermography (IRT), hyperspectral imaging, ultraviolet imaging or rgb imaging.

Infrared thermography is becoming increasingly important in non-destructive testing for applications such as subsurface defect detection [2–4]. A typical IRT setup contains a modulated heat source and a thermal camera that can record the heating or cooling curve

of each pixel in the detector array. Based on these temperature differences, it is possible to obtain information about the subsurface material properties, since heat dissipation depends not only on the surface but also on the subsurface material properties.

Another NDT camera technique is hyperspectral imaging (HSI). In this technique, we record the reflectance intensity of hundreds of narrow wavelength bands for each pixel. This measurement technique allows us to obtain more information about the chemical composition of the scanned object. HSI is commonly used in agriculture [5,6] (to quantify plant health), forensics [7,8], drug detection [9,10], NDT [11,12] and remote sensing [13]. Most hyperspectral imaging techniques are based on line-scan measurements. Recently, snapshot hyperspectral cameras developed by IMEC [14,15] have seen a resurgence. These cameras are capable of operating as full area cameras instead of line-scan cameras.

An additional NDT method involves employing ultraviolet (UV) cameras. These cameras operate in the ultraviolet region of the electromagnetic spectrum. This UV range includes wavelengths between 100 and 400 nm. An ultraviolet light source is used to illuminate the object, creating fluorescence in the object that can be measured with a detector that is sensitive in the UV range. Typical applications for these cameras are forensics (fingerprint and bodily fluid detection) [16,17], painting inspection [18], emission monitoring [19] and coating inspection [20]. Compared to light in the visible spectrum, UV light has a larger diffraction limit, making it possible to distinguish smaller objects or defects than when using visible light [21]. In this article, we use a subpart of the UV range, namely the UV-A, spanning from 320 nm to 400 nm.

When using camera-based techniques, objects are usually examined from a stationary position, which limits the size and/or resolution of the samples that can be examined. Larger samples require a greater distance between the camera and the sample to examine the entire object. Consequently, the resolution of the captured images is lower in a given area. Small defects may not be visible due to insufficient resolution. For example, in thermography, the rule of thumb is that the diameter of a defect must be at least the area of $3 \times 3$ pixels [22]. Dynamic line thermography (DLST) is often used when inspecting large surfaces or surfaces in a continuous process such as cold-rolled steel fabrication [22]. This technique uses a linear heat source and a thermal imaging camera at a specific distance from that heat source. The object of interest moves relatively in a linear motion using a traverse system or conveyor belt. The resulting images are reconstructed from the sequence of captured images.

This reconstruction is called transforming a spatial matrix into a temporal matrix. The spatial matrix is a three-dimensional construct in which raw images from a camera are sequentially stacked along the z-axis. This stacking process occurs as the object moves through the camera's field of view, essentially capturing a spatial progression of the object's movement. Imagine it as layering snapshots in a vertical sequence, where each layer represents the object at a different point in its journey across the camera's view. In contrast, the temporal matrix is derived from the spatial matrix but with a crucial difference: the object remains stationary in the images. This creates a scenario where time is the varying factor, not the position of the object. The images in the temporal matrix represent different moments but from the perspective of the object's fixed location. Essentially, while the spatial matrix captures movement through space, the temporal matrix captures changes over time, with the object's position held constant.

The disadvantage of this method is that the acquisition speed of the camera must be synchronized with the translation speed of the object. Otherwise, the reconstruction will not be possible. With the algorithm proposed in this article, it is possible to obtain a correct reconstruction of the object even if the acquisition speed of the camera and the translation speed are not synchronized. In a production environment, changing the camera acquisition speed or the linear motion is not always possible.

When creating reconstructed images in thermography and hyperspectral inspections, the motion of the sample and the frame rate of the camera are synchronized. In this way, it is relatively easy to extract the desired area from each frame of the recorded sequence,

as described in [23]. In industry, however, it is rarely possible to match the movement speed of the object to the frame rate of a camera, as this would result in fewer objects being produced per hour. Cameras, on the other hand, often have a fixed frame rate, making it impossible to adjust the speed at which an object moves.

In this manuscript, a novel algorithm is proposed for reconstructing images acquired with an unsynchronized dynamic measurement. The current literature features several papers on dynamic line-scanning thermography, yet our work introduces a novel aspect. Our method uniquely allows the camera and object speed to operate asynchronously, eliminating the need for upfront synchronization. The process is facilitated by our innovative algorithm, a technique not previously explored according to our knowledge. This advancement significantly contributes to the field by offering enhanced flexibility and efficiency in thermographic imaging. This algorithm can be used for dynamic line-scan thermography measurements, enabling us to detect defects in objects that are hidden beneath the surface, while the object is moving. Essential for an industrial process. This algorithm also provides a solution for measurements with a heterogeneous illumination source, as is often the case when using HSI or UV cameras. An illumination source is difficult to obtain in industrial environments due to safety or financial reasons. Essentially, it employs the same algorithm and reconstructs an image of a specific spatial location that has an even illumination, thus facilitating defect detection.

It is possible to image samples in multiple sections and then stitch them together to examine the entire sample with sufficient resolution. This method only works if the light source used is homogeneous. The slightest difference in illumination will cause a deviation in the recorded data, making it pointless to stitch the different sections together. This is illustrated in Figure 1, where the individual images are stitched together to reconstruct the original object.

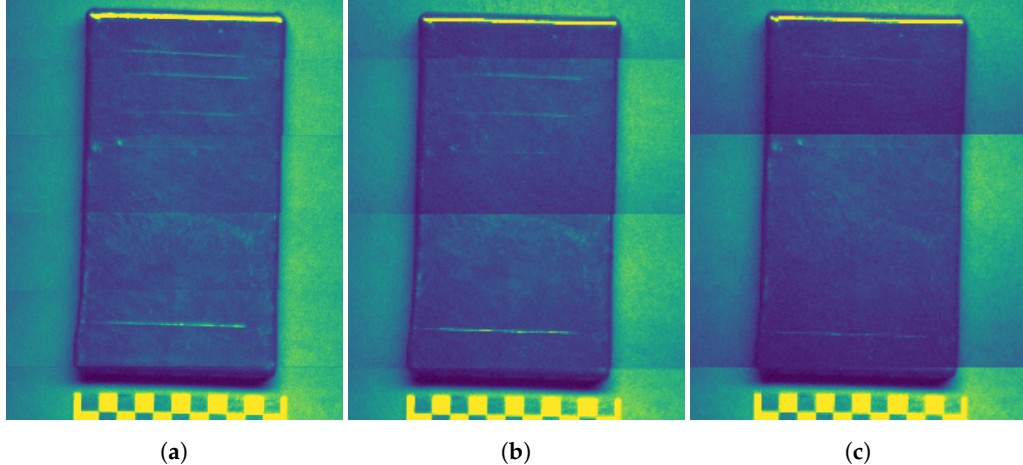

(**a**)             (**b**)             (**c**)

**Figure 1.** Three UV images (**a**–**c**) that are composed of separate images, taken at different locations for the same sample. (**a**) Six separate images; (**b**) four separate images; (**c**) three separate images. The stitching lines are clearly visible, and this could result in misclassifications in defect detection.

In this article, we first explain in detail the algorithm used, followed by a description of the equipment used for the measurements. In Section 3, the algorithm is applied to three different camera technologies: UV, IR and RGB, demonstrating the usefulness of this novel algorithm.

## 2. Materials and Methods

To evaluate the applications and show the robustness of the proposed algorithm, different camera types are used. Each camera operates at a different wavelength and has a different frame rate. The cameras used in this paper are:

- UV Camera: IMPERX GEV B1620M, frame rate: 35 fps
- Hyperspectral Camera: Photonfocus MV1-D2048x1088-HS02-96-G2, frame rate: 42 fps

- Infrared Camera: Flir A715, frame rate: 30 fps

The objects of interest were placed on a translation stage that moved with a fixed speed of 15 mm/s. The cameras were placed above the translation stage and pointed downward, perpendicular to the translation. A corresponding light source (heating source for IR) provided continuous illumination during the experiments. An image of the setup can be seen in Figure 2. An enclosure was made to block harmful UV radiation during the measurement.

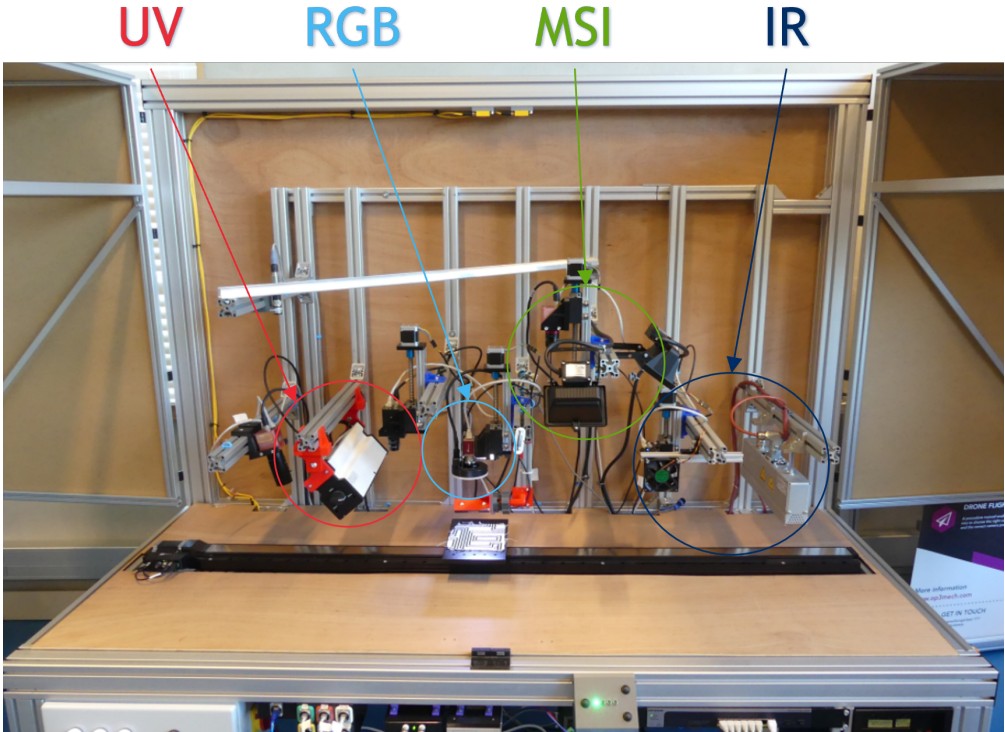

**Figure 2.** Image of the experimental setup with the housing open. Multiple cameras and corresponding light sources are mounted above a translation stage to allow examination of a sample in multiple wavelengths.

### 2.1. Algorithm

The novel stacking algorithm is developed to correct image sequences where the speed of the camera and the moving object are not synchronized, provided that the object maintains a constant linear motion and the distance between the camera and the object is constant. Initially, the algorithm captures raw images during the object's movement, ensuring complete recording. This is followed by preprocessing the data, which varies depending on the camera type. The crucial step involves calculating the pixel shift between frames, which can be achieved through two methods: tracking a checkerboard pattern or using known camera parameters. Once the pixel shift is determined, it is converted into a fraction with a maximum denominator of 10 to balance computational efficiency and accuracy. The images are then upscaled in length according to this denominator. The next phase involves creating a temporal matrix by iterating over frames and copying corresponding rows from the original sequence into this matrix. After constructing the temporal matrix, any shifts due to the object's motion are corrected. Finally, the algorithm downscales the temporal matrix in length by the same fraction's denominator to complete the correction process. This series of steps ensures that the final image sequence accurately represents the object's motion relative to the camera's position.

Figure 3 shows a visual overview of the different time steps of the stacking algorithm. The list below explains the algorithm in more detail.

Several steps are required to correct an image sequence where the camera and object speed is not matched. However, the object has to move in a constant linear motion. For an overview of the correction algorithm, see Figure 3. The following list explains the algorithm in detail.

1.  Capture raw data
    The images are captured during translation of the object while making sure that the start and end of the object are fully recorded. This is displayed graphically in Figure 4 in rows 1–2.

2.  Preprocess data
    Depending on the camera type, preprocessing is required, e.g., NUC (non-uniformity correction) for IR cameras or demosaicing for snapshot HSI cameras.

3.  Find pixel shift
    To calculate the pixel shift from frame n to n + 1, we present two methods. The first method is to detect and track a checkerboard in a sequence of frames. The upper left corner of the checkerboard is tracked across a series of frames and fitted with a first-degree function to minimize the error. The other method is to calculate the pixel shift. This is possible if the camera parameters are known. For more details, see Section 2.1.1 [24].

4.  Factorize pixel shift
    The pixel shift is converted to a fraction with a maximum denominator of 10. The maximum denominator is chosen empirically to limit the computational cost while still providing an accurate result. One should consider the necessary accuracy needed for the application versus the computational cost.

5.  Upscale the images
    The preprocessed images are resized in length, through interpolation, by a factor equal to the denominator of the previously calculated fraction.

6.  Create a temporal matrix
    This step consists of a main loop and a nested loop. The main loop iterates over all the different frames of the temporal matrix. The number of frames is calculated by $ft = l * Fraction$, where $ft$ is the resulting number of frames in the temporal matrix and $l$ is the length of a single frame from the original sequence. The nested loop iterates over the original sequence and copies the corresponding rows from the original sequence into the temporal matrix. This step is shown below in pseudocode Algorithm 1:

---

**Algorithm 1** Creating the temporal matrix

---

**for** $i \leftarrow 1, ft$ **do**
    **for** $j \leftarrow 1, fo$ **do**
        $T_{Mat} \leftarrow O_{Mat}$
        Where $T_{Mat}$ = line: $j * N : (j * N) + N$ from frame $k$
        Where $U_{Mat}$ = line: $(i * N) + N : i * N$ from frame $j$
    **end for**
**end for**

---

with $fo$ as the amount of original frames, $T_{Mat}$ as the temporal matrix, $U_{Mat}$ as the upscaled sequence, $N$ as the numerator, $O_{Mat}$ as the original measured matrix and $D$ as the denominator of the fraction pixel shift. This operation is displayed in rows 5–6 of Figure 4.

7.  Correct temporal matrix
    The temporal matrix indicates a shift of the object in the direction of translation. This can be corrected by shifting the image.

8.  Downscale the images
    The resulting temporal matrix is scaled down in length by a factor equal to the denominator of the previously calculated fraction.

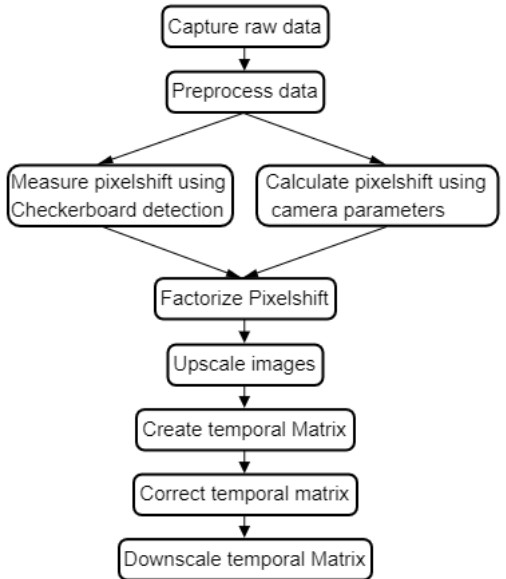

**Figure 3.** Flowchart of the different steps in the proposed algorithm.

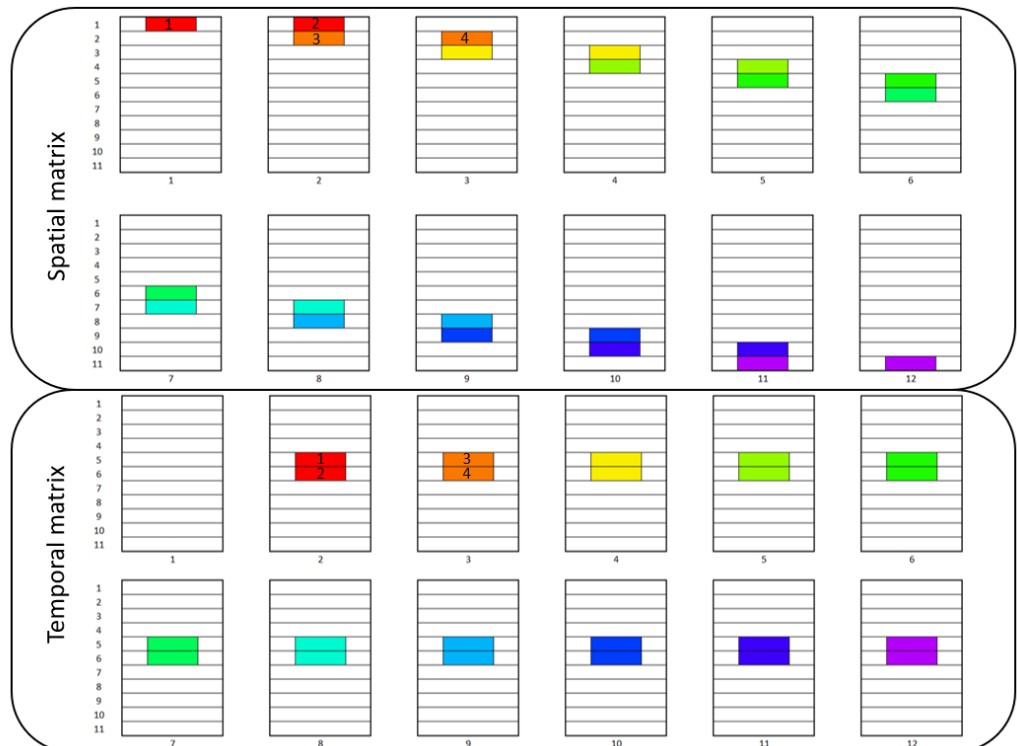

**Figure 4.** Transformation of the spatial matrix of the raw images $O_{mat}$ into a temporal matrix $T_{mat}$. The first two rows of the images show the spatial matrix $O_{mat}$, and the last two rows show the stitched temporal matrix $T_{mat}$. The colors correspond to the moment after passing the light source. The color red corresponds to the spatial matrix directly after passing the light source. Matching colors are combined in the same image from the temporal matrix. For the first five reconstructed sections, these are numbered, meaning that number 1 in the spatial matrix will be located at number 1 in the temporal matrix. The numbers below the matrix represent the frame number and the numbers on the right indicate the line number of a single frame.

### 2.1.1. Calculating Correction Factor without the Use of a Checkerboard

The algorithm uses a checkerboard pattern to calculate the required correction factor, as explained in step 3 of the proposed algorithm in Section 2.1. However, it is also possible to determine this factor without the use of a checkerboard. The factor can be calculated using the Equations (1)–(3).

$$H = 2 * d * \frac{\frac{p}{2}}{f} \tag{1}$$

The distance (H) detected by a pixel line of a camera can be calculated using the focal length ($f$), ($p$) the pixel width and ($d$) the height of the camera above the sample. The calculation can be performed similarly using the field of view of a camera. The number of pixels by which the object is translated per second can be calculated using Equation (1) and the translation speed of the object ($v$).

$$px/sec = \frac{H}{v} \tag{2}$$

Using the pixel shift per second and the recording rate of the camera, you can calculate the pixel shift for each frame of the image sequence.

$$factor = \frac{px/sec}{framerate} \tag{3}$$

### 2.1.2. Algorithm Performance

For the stacking algorithm, a Dell XPS notebook is used with the following specs:

1. CPU: Intel i7 6 cores-2.6Ghz
2. RAM: 32 GB

The calculation for the provided examples is performed purely on CPU power; further improvements could be made through parallelisation on CPU and/or GPU. The time needed to complete the stacking algorithm is very dependent on the processing power, the desired accuracy (scale factor) and the image size. In Table 1, the processing times for each experiment can be found.

**Table 1.** The table shows the processing times of the algorithm to correctly stitch the images for further processing. The times represent the different experiments, as shown in Section 3.

|  | UV | MSI | IR |
|---|---|---|---|
| Frame size (pixels) | $1208 \times 1608$ | $408 \times 208$ | $640 \times 480$ |
| Number of frames | 498 | 710 | 1008 |
| Processing time of algorithm (s) | 434 | 89 | 101 |

### 3. Results

In this section, we showcase the ability of our algorithm to correct the images, invariant of the chosen speed of the camera and translation stage. This algorithm is able to correct the images so that a temporal matrix can be created, on which further processing can be performed. To further validate the potential of the algorithm, we apply this across various camera technologies in non-destructive testing, each offering unique insights into different imaging scenarios. Our study focuses on evaluating this algorithm using three line-scan camera techniques: IR, MSI and UV. The results from each camera type demonstrate the algorithm's proficiency in enhancing defect clarity compared to raw images, while being invariant of this enhancement across diverse imaging conditions underscores the algorithm's versatility and potential in improving image analysis in real-world applications and defect detection in non-destructive testing applications.

### 3.1. IR Camera

A commonly used inspection sample in active thermography is a flat bottom-hole plate, as shown in Figure 5. The air trapped in the voids cools differently than the surrounding material. As heat is trapped, the air remains at a higher temperature and therefore shows up as a hotspot in the thermograms. Depending on the thermal properties of the sample material and the characteristics of the defects (depth, size, type of defect, . . . ), the time needed to show a hotspot in the thermogram may vary. For this reason, the spatial image sequence, as shown in Figure 6, must be converted into a temporal image sequence, using the novel algorithm from Section 2.1. In the temporal data set, as shown in Figure 7, the sample under investigation remains at the same location and the cooling can be investigated. It is noticeable that the air pockets are smaller when the location from line source is further from the heating source. From Figure 6, it is noticeable that due to the angle of the camera, the edge of the object appears smaller than the same edge in image c. This effect only appears in the measured data. When the algorithm is applied to the measured data, a single location is used to reconstruct the object. Because the same location is used to reconstruct the image, there is no noticeable deformation of the object, as can be seen in the temporal matrix in Figure 7.

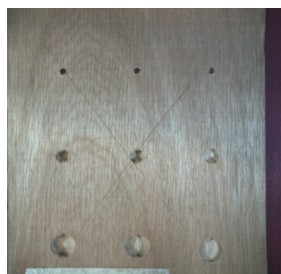

**Figure 5.** RGB image of the inspected bottom wooden sample, so the defects are visible. The dimensions are the following: $160 \times 100 \times 5.8$ mm.

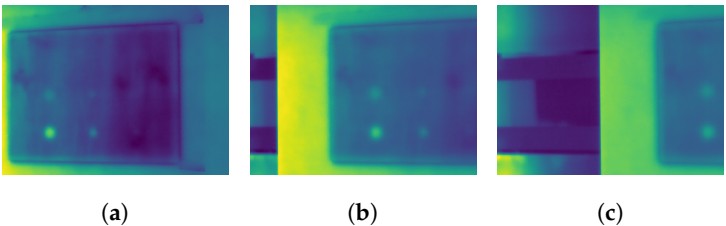

|     (a)     |     (b)     |     (c)     |

**Figure 6.** Three frames from the sequence captured by the IR camera while moving the object. (**a**) Frame 300; (**b**) frame 550; (**c**) frame 800. The frames correspond to an number of seconds after passing the heating source. (**a**) 10 s; (**b**) 18.3 s; (**c**) 26.67 s.

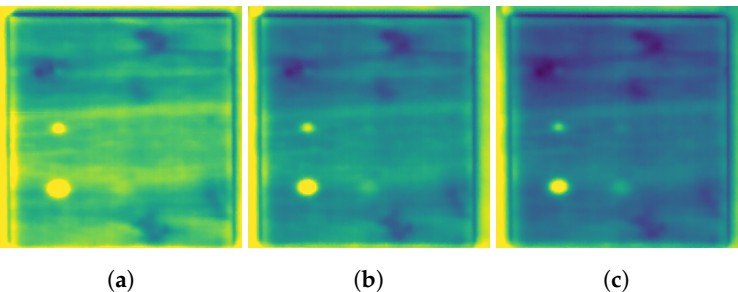

|     (a)     |     (b)     |     (c)     |

**Figure 7.** Three frames from the temporal matrix, as created by the novel algorithm. (**a**) Frame 1; (**b**) frame 428; (**c**) frame 856. The ratio of width/height of the object matches the pixel dimensions of the reconstructed image, confirming the accuracy of the stitching algorithm. The air pockets are visible as circular hot spots. A less accurate stitching would result in elliptical hot spots.

### 3.2. UV Camera

UV cameras can be used to detect scratches with higher contrast than with normal RGB cameras. However, an additional UV light source is required. These light sources produce a very narrow beam of light and therefore create an uneven illumination in the image, as can be seen in Figure 8. This algorithm not only facilitates measurements with heterogeneous illumination sources, which is a common challenge with hyperspectral imaging (HSI) or UV cameras, but also provides a solution that is particularly useful in industrial settings. Achieving even illumination can be difficult due to safety or financial constraints in such environments. By applying the same algorithm to reconstruct an image at a specific spatial location, it ensures even illumination, thereby enhancing the detection of defects. Selecting a single line position can address this issue. However, since the starting position is initially unknown, imaging the entire sample becomes necessary. Our innovative algorithm plays a crucial role here, enabling us to seamlessly stitch together the images. This process allows us to reconstruct the image at a specific, targeted location, effectively overcoming the challenge of the unknown initial position. The sample studied, which was used for this measurement, is an epoxy-coated carbon steel sample. Several incisions of varying depth were made in this sample using a knife blade. An RGB image of this sample can be seen in Figure 9. From the RGB image, it is difficult to see all the different incisions. When looking at the UV sequence in Figure 8, the lowest cuts are very visible in the first image and second image (Figure 8a,b) but they become less visible in the last image (Figure 8c).

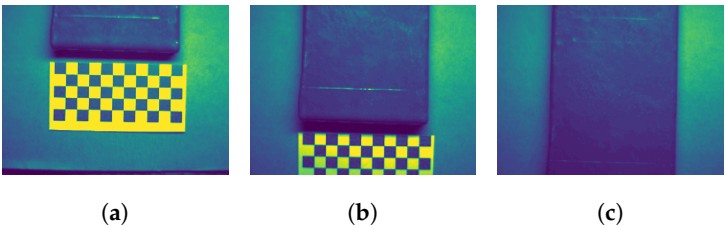

(**a**)  (**b**)  (**c**)

**Figure 8.** Three frames from the spatial matrix: (**a**) frame 90 (2.57 s); (**b**) frame 150 (4.29 s); (**c**) frame 230 (6.57 s).



**Figure 9.** RGB image of the epoxy-coated steel sample with incisions. The incisions are marked with red squares. The sample has the following dimensions: $100 \times 50 \times 13$ mm.

When the spatial matrix is converted to a temporal matrix, the object under study is recreated. Three separate frames from the spatial matrix can be seen in Figure 10. Each frame in the spatial matrix represents a different physical location and, in this case, a different intensity of the light source used. Figure 10a shows that on the first image, the incisions are most obvious, but other artifacts enter the image due to the reflection of the UV light. Figure 10b is better suited to analyze this type of defect. In Figure 10c, we notice a significantly darker image. This darkness is attributed to the algorithm reconstructing the

image at a spatial position with less intense illumination, resulting in a darker appearance in the reconstructed output.

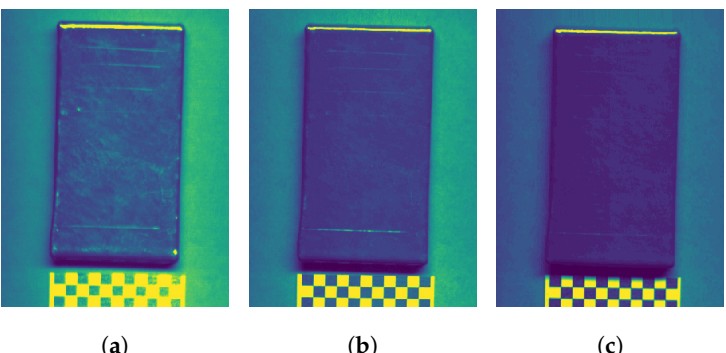

|     |     |     |
| --- | --- | --- |
| (**a**) | (**b**) | (**c**) |

**Figure 10.** Three frames from the temporal matrix: (**a**) frame 1; (**b**) frame 50; (**c**) frame 100. Notice that due to heterogeneous illumination from the UV light source, the different frames of the temporal matrix show a different brightness value.

*3.3. HSI Camera*

When using hyperspectral cameras, an additional dimension (wavelength) is added to the processing. Just as with UV lamps, uneven illumination presents a challenge in hyperspectral imaging. To address this, we can apply the same approach using our innovative algorithm, allowing us to concentrate on a specific line. This technique effectively mitigates the issue of uneven lighting, enhancing the accuracy and quality of hyperspectral imaging results. In Figure 11, the HS camera displays three distinct bands or wavelengths, illustrating the presence of uneven illumination across the image. This is particularly noticeable as the bottom part of the image appears darker. This variation in brightness across different wavelengths not only highlights the challenges in maintaining consistent illumination in hyperspectral imaging but also underscores the importance of advanced processing techniques to compensate for these inconsistencies, ensuring that the final image accurately represents the subject under observation. The object shown is a piece of fabric with weaving imperfections and a black and white woven pattern. This pattern makes it difficult to examine with an RGB camera, as shown in Figure 12. When selecting a specific band, the woven black and white pattern will not be visible and the weaving will be clearer.

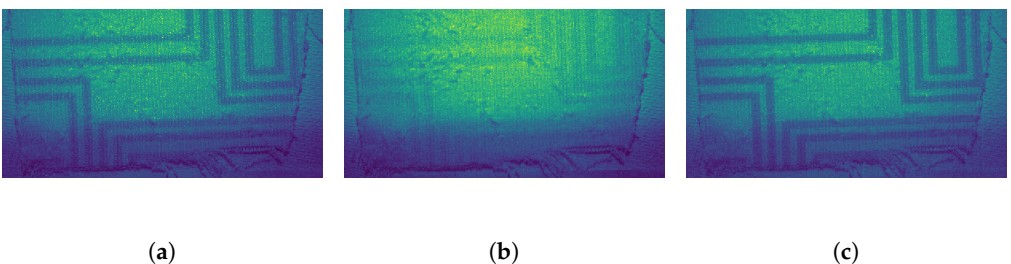

|     |     |     |
| --- | --- | --- |
| (**a**) | (**b**) | (**c**) |

**Figure 11.** Three separate bands from the HS camera for a single frame in the sequence, before applying the novel algorithm. The identification of weaving errors is complicated through the heterogeneous illumination. (**a**) Wavelength: 677 nm; (**b**) wavelength: 776 nm; (**c**) wavelength: 959 nm.

Figure 13 shows the temporal matrix for a single hyperspectral band. It can be seen that not only the choice of wavelength is important but also the temporal position to obtain a high contrast between the weaving defects and the normal tissue. In Figure 13c, for example, the illumination of this spatial position is too large, causing the object to be over-saturated.

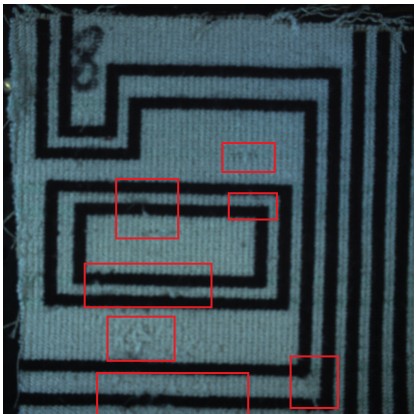

**Figure 12.** RGB image of the textile sample with defects. The defects are marked with red squares. The dimensions of the sample are 200 × 200 × 4 mm.

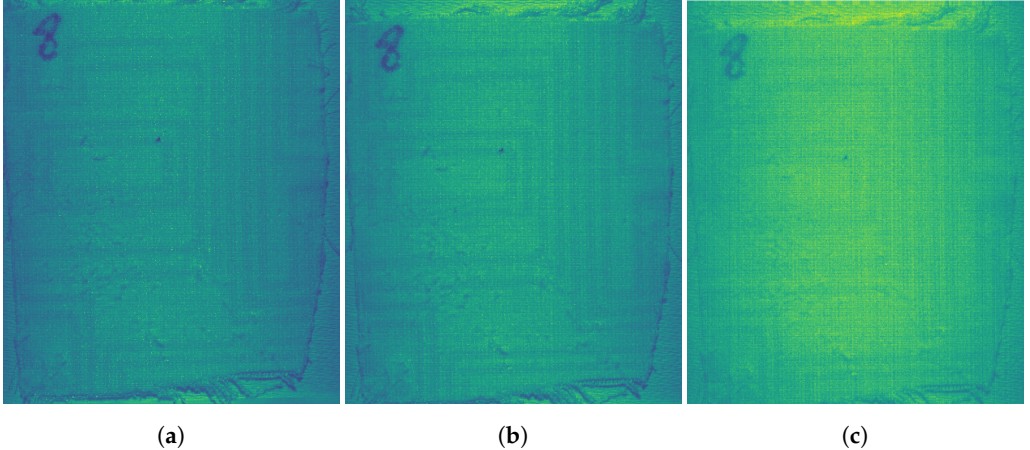

(**a**)           (**b**)           (**c**)

**Figure 13.** Three frames from the temporal matrix for a single wavelength (776 nm). Notice that these images display a homogenous illumination, which improves the detection of weaving defects. (**a**) Frame 1; (**b**) frame 107; (**c**) frame 215.

## 4. Discussion and Conclusions

The algorithm discussed in this manuscript allows industry to perform non-destructive inspections during manufacturing without having to adjust the translation speed of the objects to the frame rate of the camera. Nowadays, it is necessary to synchronize the translation speed of an object with the acquisition rate of the camera used. The frame rate of a camera is often fixed; consequently, the production process has to be adapted to the capture rate. With this algorithm, the production process is no longer dependent on the camera used. In other words, the number of objects produced is not determined by quality control. The results show that we are able to successfully apply the algorithm to three different cameras, each camera operating a different wavelength. We are able to achieve an accuracy of 0.1 pixels while stitching the temporal matrix. This accuracy can be further increased by changing the upscaling factor if this is necessary for the specific purpose. The proposed algorithm can measure the pixel shift in two different ways, through checkerboard detection or field-of-view calculation.

In addition to the aforementioned advantage of this algorithm, it also allows samples to be examined without the adverse effects of a heterogeneous light source.

## 5. Future Work

In future work, the redesigned matrix can be used to perform automatic inspections using artificial intelligence. The processing time is very dependent on the PC used, the desired accuracy and the size of the images. Additional gains can be found through parallel computing and/or GPU implementation. This will be handled in future work. Also,

accuracy calculations can be carried out, determining how the different parameters of the algorithm, such as ratio camera speed, translational speed and upscaling factor, have an influence on the reconstruction accuracy.

**Author Contributions:** Conceptualization, S.V. (Simon Verspeek), T.D.K. and B.R.; methodology, S.V. (Simon Verspeek), T.D.K. and B.R.; software, S.V. (Simon Verspeek) and T.D.K.; validation, S.V. (Simon Verspeek) and T.D.K.; resources, S.V. (Steve Vanlanduit) and G.S.; writing—original draft preparation, S.V. (Simon Verspeek) and T.D.K.; writing—review and editing, B.R., X.M., S.V. (Steve Vanlanduit) and G.S.; supervision X.M., S.V. (Steve Vanlanduit) and G.S.; project administration, S.V. (Steve Vanlanduit) and G.S.; funding acquisition, S.V. (Steve Vanlanduit) and G.S. All authors have read and agreed to the published version of the manuscript.

**Funding:** This research has been funded by Research Foundation-Flanders under grant Doctoral (PhD) grant strategic basic research (SB) 1SC0819N (Simon Verspeek) and SPF Economy ETF—PhairywinD project.

**Institutional Review Board Statement:** Not applicable.

**Informed Consent Statement:** Not applicable.

**Data Availability Statement:** The data that support the findings of this study are openly available at 10.5281/zenodo.10433373.

**Conflicts of Interest:** The authors declare no conflict of interest.

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
