# Peer review of "A Novel Line-Scan Algorithm for Unsynchronised Dynamic Measurements"

_applsci, doi:10.3390/app14010235_

Round 1

Reviewer 1 Report

Comments and Suggestions for Authors

This paper developed a line scan algorithm so that images captured at different timestamp can be stitched together regardless of whether the imaging frame rate is consistent with the objective translating rate. I found the paper to be overall well-written and well-structured. However, some of the important information is missing, which leaves some questions and confusions. More importantly, the novelty of this work is not clear regarding how this approach can be helpful in improving the defect diagnosis process. Furthermore, in the "Results" section, I didn't find the proposed approach was applied on all the cases. Therefore, the current form of the manuscript is not up to the standard of this journal.

Major comments:       

1. Page 2, line 69, "...Figure 3..." The purpose of figure 3 here is not clear since this paper is not about this approach at all, nor does it help readers understand the process.

2. Page 7, section 2.1., not sure if the proposed approach is mentioned in this section.

3. Page 9, figure 12., not sure if the proposed approach is applied to these images.

4. Page 12, line 234, "...0.1 pixel...," where does this number come from?

Minor comments:

1.  Page 2, line 69, "...Figure 3..." the figure index should start from 1 since this is the first figure mentioned in this paper.

2.  Page 2, line 70, "...synchronized...," what do you mean by "synchronized" here? Do you mean the frame rate of the camera be the same as the translation speed of the objective?

3. Page 5, the details about the algorithm. Any schematic of the approach with figures and annotations rather than just text description would be helpful. 

4. Page 7, figure 6, due to the angle of the camera toward the objective, the left edge of the object on (a) seems to be shorter than that in (c). How do you resolve that, by some sort of calibration?

5. Page 8, in the caption of figure 7, "...it is apparent that..." what do you mean by this sentence?

6. Page 8, line 198, "These light sources...," not sure what does this sentence mean.

7. Page 9, figure 10, any comment on (c)?

8. Page 9, line 214, again, the figure index should follow the consecutive order. After figure 10, should be figure 11 rather than figure 12.

Author Response

Reviewer comment: “However, some of the important information is missing, which leaves some questions and confusions.”

Author reply

In the revised paper, we have added several paragraphs to elaborate more on the different aspects for which information was missing (see the replies to the following questions).

 Reviewer comment: “More importantly, the novelty of this work is not clear regarding how this approach can be helpful in improving the defect diagnosis process.”

Author reply

We have added a paragraph to further discuss the novelty of the work and in particular how it can be helpful in improving the defect diagnosis process (see Page 2 of the revised manuscript):

In this manuscript, a novel algorithm is proposed for reconstructing images acquired with an unsynchronized dynamic measurement. The current literature features several papers on dynamic line scanning thermography, yet our work introduces a novel aspect. Our method uniquely allows the camera and object speed to operate asynchronously, eliminating the need for upfront synchronization. The process is facilitated by our innovative algorithm, a technique not previously explored according to our knowledge. This advancement significantly contributes to the field by offering enhanced flexibility and efficiency in thermographic imaging. This algorithm can be used for dynamic line scan thermography measurements, enabling us to detect defects in objects that are hidden beneath the surface, while the object is moving. Essential for an industrial process. This algorithm also provides a solution for measurements with a heterogeneous illumination source, as is often the case when using HSI or UV cameras. As an even illumination source is difficult to obtain in industrial environments due to safety or financial reasons. Essentially employing the same algorithm and reconstructing an image of a specific spatial location that has an even illumination. Thus facilitating the defect detection.

Reviewer comment: “Furthermore, in the "Results" section, I didn't find the proposed approach was applied on all the cases..”

Author reply

In the revised paper we have clarified on which cases the approach was used (see Page 7 of the revised manuscript):

In this section, we showcase the ability of our algorithm to correct the images, invariant of the chosen speed of the camera and translation stage. This algorithm is able to correct the images, so that a temporal matrix can be created, on which further processing can be done.

Also for each experiment in the results section we have clarified that the algorithm used in this paper, has been used to create the temporal matrix:

UV:
Selecting a single line position can address this issue. However, since the starting position is initially unknown, imaging the entire sample becomes necessary. Our innovative algorithm plays a crucial role here, enabling us to seamlessly stitch together the images. This process allows us to reconstruct the image at a specific, targeted location, effectively overcoming the challenge of the unknown initial position.

HSI:
Just as with UV lamps, uneven illumination presents a challenge in hyperspectral imaging. To address this, we can apply the same approach using our innovative algorithm, allowing us to concentrate on a specific line. This technique effectively mitigates the issue of uneven lighting, enhancing the accuracy and quality of hyperspectral imaging results.

Reviewer comment: “Page 2, line 69, "...Figure 3..." The purpose of figure 3 here is not clear since this paper is not about this approach at all, nor does it help readers understand the process.”

Author reply

The caption of Figure 3 has been modified and the figure has been moved to a section after the algorithm. The terminology is also adapted to match the terminology used in the algorithm section of 1.1 on page 4. To further improve clarity, we have added an additional paragraph in Section 1.1 to provide an overview of the algorithm. See comment 1 of Reviewer 2, or the added paragraph on Page 5.

Reviewer comment: “Page 7, section 2.1., not sure if the proposed approach is mentioned in this section.”

Author reply

We have modified the section to better reflect the proposed approach (see Page 8 of the revised manuscript):

For this reason, the spatial image sequence, as shown in Figure 6 must be converted into a temporal image sequence, using the novel algorithm from Section 1.1.

And the caption of figure 7 is also modified to:

Three frames from the temporal matrix, as created by the novel algorithm

Reviewer comment: “Page 9, figure 12., not sure if the proposed approach is applied to these images.”

Author reply

We have clarified further how the approach was applied to these images in the text and the figure caption (see Page 10 of the revised manuscript):

Change in the caption of the figure:
Three separate bands from the HS camera for a single frame in the sequence, before applying the novel algorithm.

Modification in the paragraph:
In Figure 9, the HS camera displays three distinct bands or wavelengths, illustrating the presence of uneven illumination across the image. This is particularly noticeable as the bottom part of the image appears darker. This variation in brightness across different wavelengths not only highlights the challenges in maintaining consistent illumination in hyperspectral imaging but also underscores the importance of advanced processing techniques to compensate for these inconsistencies, ensuring that the final image accurately represents the subject under observation.

Reviewer comment: “Page 12, line 234, "...0.1 pixel...," where does this number come from?”

Author reply

The statement of accuracy of 0.1 pixel has been removed from the discussion section, as this number could not be verified during the measurements. However, an additional section with future work has been added that addresses the reconstruction accuracy (see Page 12 of the revised manuscript):

In future work, the redesigned matrix can be used to perform automatic inspections using artificial intelligence. The processing time is very dependent on the used PC, the desired accuracy and the size of the images. Additional gains can be found through parallel computing and/or GPU implementation. This will be handled in future work. Also, accuracy calculations can be carried out, determining how the different parameters of the algorithm such as ratio camera speed and translational speed, upscaling factor have an influence on the reconstruction accuracy.

Reviewer comment: “Page 2, line 69, "...Figure 3..." the figure index should start from 1 since this is the first figure mentioned in this paper.”

Author reply

We have changed the figure numbering (see Page 2 of the revised manuscript).

Reviewer comment: “Page 5, the details about the algorithm. Any schematic of the approach with figures and annotations rather than just text description would be helpful.”

Author reply

We have modified the position and caption of Figure 3 to make it more clear. Now the terminology of the caption matches the terminology of the algorithm, thus improving clarity. In the textual description of the algorithm, we have included references to the corresponding figures where relevant, ensuring a clearer understanding and visualization of the algorithm's steps and functionalities. Also see reply to comment 4.

Modified caption of the figure:
Transformation of the spatial matrix of the raw images Omat into a temporal matrix Tmat. The first two rows of the images show the raw image sequence Omat, the middle two rows visualize the division into parts corresponding to the pixel shift between successive images, and the last two rows show the stitched temporal matrix Tmat. The colors correspond to the moment after passing the light source. The color red corresponds to the spatial directly after passing the light source. Matching colors are combined in the same image from the temporal matrix.

Reviewer comment: “Page 7, figure 6, due to the angle of the camera toward the objective, the left edge of the object on (a) seems to be shorter than that in (c). How do you resolve that, by some sort of calibration?”

Author reply

We have added a short paragraph to explain the difference between the edge distance in (a) and (c)  (see Page 8 of the revised manuscript):

From Figure 6 it is noticeable that due to the angle of the camera, the edge of the object appears smaller than the same edge in image (c). This effect only appears on the  measured data. When the algorithm is applied to the measured data, a single location is used to reconstruct the object. Because the same location is used to reconstruct the image, there is no noticeable deformation of the object, as can be seen in the temporal matrix in Figure 7.

Reviewer comment: “Page 8, in the caption of figure 7, "...it is apparent that..." what do you mean by this sentence?”

Author reply

We have modified the sentence to clarify the meaning of ‘it is apparent’ (see Page 9 of the revised manuscript):

The figure clearly shows that the dimensions of the object match its actual physical dimensions.

Reviewer comment: “Page 8, line 198, "These light sources...," not sure what does this sentence mean.”

Author reply

The sentence has been modified to: (see Page 9 of the revised manuscript):

These light sources produce a very narrow beam of light and therefore create an uneven illumination in the image, as can be seen in the figure.

Reviewer comment: “Page 9, figure 10, any comment on (c)?”

Author reply

We have added a comment on Figure 10 (c) (see Page 10 of the revised manuscript):

In Figure 10c, we notice a significantly darker image. This darkness is attributed to the algorithm reconstructing the image at a spatial position with less intense illumination, resulting in a darker appearance in the reconstructed output.

Reviewer comment: “Page 9, line 214, again, the figure index should follow the consecutive order. After figure 10, should be figure 11 rather than figure 12.”

Author reply

We have changed the figure numbering to make the references appear in increasing order (see Page 10 of the revised manuscript).

Reviewer 2 Report

Comments and Suggestions for Authors

In my opinion, the paper includes some novelty. But, I have following comments :

-       Please give a more detailed explanation of the correction algorithm and comment on the effect of distance between the camera and the sample on the results.

-       No processing time is performed.

-       The results description is very poor, the idea is completely uncorrelated with the way the results were discussed.

Author Response

Reviewer comment: “Please give a more detailed explanation of the correction algorithm and comment on the effect of distance between the camera and the sample on the results.”

Author reply

We have added a new paragraph to give a more detailed explanation of the correction algorithm and we have commented on the effect of the distance between the camera and the sample on the results (see Page 4 of the revised manuscript):

The novel stacking algorithm is developed to correct image sequences where the speed of the camera and the moving object are not synchronized, provided that the object maintains a constant linear motion and the distance between the camera and object is constant. Initially, the algorithm captures raw images during the object's movement, ensuring complete recording. This is followed by preprocessing the data, which varies depending on the camera type. The crucial step involves calculating the pixel shift between frames, which can be achieved through two methods: tracking a checkerboard pattern or using known camera parameters. Once the pixel shift is determined, it's converted into a fraction with a maximum denominator of 10 to balance computational efficiency and accuracy.

The images are then upscaled in length according to this denominator. The next phase involves creating a temporal matrix by iterating over frames and copying corresponding rows from the original sequence into this matrix. After constructing the temporal matrix, any shifts due to the object's motion are corrected. Finally, the algorithm downscales the temporal matrix in length by the same fraction's denominator to complete the correction process. This series of steps ensures that the final image sequence accurately represents the object's motion relative to the camera's position.

 Reviewer comment: “No processing time is performed.”

Author reply

In response to the reviewer's request for clarification of the processing time, we have revised the presentation of Table 1 to enhance its clarity. (see Page 7 of the revised manuscript):

Reviewer comment: “The results description is very poor, the idea is completely uncorrelated with the way the results were discussed.”

Author reply

We have completely re-written the discussion of the results in order to better correlate with the idea of the paper (see Page 7 of the revised manuscript):

In this section, we showcase the ability of our algorithm to correct the images, invariant of the chosen speed of the camera and translation stage. This algorithm is able to correct the images, so that a temporal matrix can be created, on which further processing can be done. To further validate the potential of the algorithm, we apply this across various camera technologies in non-destructive testing, each offering unique insights into different imaging scenarios. Our study focuses on evaluating this algorithm using three line scan camera techniques: IR, MSI, and UV. The results from each camera type demonstrate the algorithm's proficiency in enhancing defect clarity compared to raw images, while being invariant of This enhancement across diverse imaging conditions, underscores the algorithm's versatility and potential in improving image analysis in real world applications and defect detection in non-destructive testing applications.              

Round 2

Reviewer 1 Report

Comments and Suggestions for Authors

I am still not convinced of the novelty of this work and the connections between the results and discussions. Some of my confusions were not addressed in the revision, which seems to add more confusions. The claimed "novel" approach was not clearly explained and the results didn't seem to correlate well with the discussions. Please see the comments below:

1. Page 3, line 90, "...defects...hidden beneath the surface..." was the approach, by which you detected the defects, discussed in the later section? How did you identify the defects underneath the surface with your surface imaging techniques? By some sort correlation between the surface imaging with in situ x-ray imaging or with ex situ material characterization?

2. Page 4, line 132, the term "temporal matrix" was used many times without explaining what it is and what it looks like. It's hard for readers to understand the necessity of converting the raw images into temporal matrix. I originally thought the temporal matrix was a necessary element to stitch together images from different time steps. However, on second reviewing of the paper, I found that might not be the case.

3. Page 4, line 146, "...in Figure 4," I expected figure 4 to show how the raw data were captured based on the title of this subsection. It turned out that Figure 4 was irrelevant to that.

4. I had a hard time understanding figure 4. What's the role of figure 4 in the paper? What does the color mean? What does the number in the vertical and horizontal axis mean? 

5. Page 9, caption of figure 7, "...created by the novel algorithm." Figure 7 looks like a cropped version of Figure 6 rather than a stitched picture to me. 

6. Page 9, caption of figure 7, "...match...," how did you know that? Did you do camera calibrations to map the pixel to physical length and compare that with the measured object dimension? If so a scale bar should be included in the figure.

7. Page 9, section 2.2., I still don't see how the approach is applied in this section. Would it be possible to show the data/image before and after the approach is applied to show the difference?

8. Page 10, line 268, "...wavelength..." how did you select those wavelengths? Did you add a filter in front of your HSI camera to select these wavelengths?

Comments on the Quality of English Language

Page 3, line 93-96. There are a lot of incomplete sentences. 

Author Response

Dear reviewer, thank you for the comments. You can find the answers in blue.

  1. Page 3, line 90, "...defects...hidden beneath the surface..." was the approach, by which you detected the defects, discussed in the later section? How did you identify the defects underneath the surface with your surface imaging techniques? By some sort correlation between the surface imaging with in situ x-ray imaging or with ex situ material characterization?

A flat bottom hole plate was used, as is customary for infrared thermography methods. This is an object with visibly present air pockets, on the bottom side of the object. On the top side, these defects are not visible. We can see the subsurface defects because a heat source is used to heat the object. Whenever there is a defect present, the heating profile will be different as compared to zones that are homogenous (thus without defect). It's important to note that this paper does not aim to provide a comparative analysis of different methods. Instead, the experimental results are presented solely to demonstrate the viability of the concept.

In the text following line is added: It is noticeable that the air pockets are smaller when the location from line source is further from the heating source. 

  1. Page 4, line 132, the term "temporal matrix" was used many times without explaining what it is and what it looks like. It's hard for readers to understand the necessity of converting the raw images into temporal matrix. I originally thought the temporal matrix was a necessary element to stitch together images from different time steps. However, on second reviewing of the paper, I found that might not be the case.

To correct for the misunderstanding a separate paragraph is included to address how the spatial and temporal matrix have to be interpreted in the paper:
This reconstruction is called transforming a spatial matrix into a temporal matrix. The spatial matrix is a three-dimensional construct in which raw images from a camera are sequentially stacked along the z-axis. This stacking process occurs as the object moves through the camera's field of view, essentially capturing a spatial progression of the object's movement. Imagine it as layering snapshots in a vertical sequence, where each layer represents the object at a different point in its journey across the camera's view. In contrast, the temporal matrix is derived from the spatial matrix, but with a crucial difference: the object remains stationary in the images. This creates a scenario where time is the varying factor, not the position of the object. The images in the temporal matrix represent different moments but from the perspective of the object's fixed location. Essentially, while the spatial matrix captures movement through space, the temporal matrix captures changes over time, with the object's position held

Figure 4 is also modified, making the difference between spatial and temporal matrix more clear now. Also, numbers for the first five sections that are reconstructed are included, and the caption is expanded with the following text: For the first five reconstructed sections these are numbered, meaning that number 1 in the spatial matrix will be located at number 1 in the temporal matrix.

  1. Page 4, line 146, "...in Figure 4," I expected figure 4 to show how the raw data were captured based on the title of this subsection. It turned out that Figure 4 was irrelevant to that.

The choice was made to not use actual raw images, instead, we used a graphical representation. This way we can make it more comprehensible using different colors, and these colors can be matched in the temporal and spatial matrix. If real images were to be used, the authors are of the impression that this will not be as clear as the schematic drawing. Also, it is mentioned very clearly in the text that it is a graphical representation of the text: This is displayed graphically in Figure 4

  1. I had a hard time understanding figure 4. What's the role of figure 4 in the paper? What does the color mean? What does the number in the vertical and horizontal axis mean? 

As mentioned in remark 2, figure 4 has been made more comprehensible through the elimination of the second row. Also, an additional caption is added to explain the numbers. See additional caption:
The numbers below the matrix represent the frame number and the numbers on the right indicate the line number of a single frame.

  1. Page 9, caption of figure 7, "...created by the novel algorithm." Figure 7 looks like a cropped version of Figure 6 rather than a stitched picture to me. 

And yet it is a reconstructed image by the algorithm. This is apparent by the size of the air pockets. In the first image, the air pockets are visible and in the subsequent images, these are less pronounced. This is because the images are stitched in a location that is further away from the heating source.

  1. Page 9, caption of figure 7, "...match...," how did you know that? Did you do camera calibrations to map the pixel to physical length and compare that with the measured object dimension? If so a scale bar should be included in the figure.

No camera calibrations were performed, however, the ratio is measured between the pixel length and pixel width of the reconstructed image. This ratio matched the measured ratio of the object. This is added to the caption:

The ratio of width/height of the object matches the pixel dimensions of the reconstructed image, confirming the accuracy of the stitching algorithm.

  1. Page 9, section 2.2., I still don't see how the approach is applied in this section. Would it be possible to show the data/image before and after the approach is applied to show the difference?

Figure 9 shows the raw images captured at different times by the camera. The caption of the image is modified to:
Three frames from the spatial matrix (raw images, before applying the algorithm) (a) Frame 90 (2.57 seconds); (b) Frame 150 (4.29 seconds); (c) Frame 230 (6.57 seconds).

 Also an additional paragraph is included in Section 2.2:
Therefore, it is not possible to obtain a single image with an even illumination where all of the defects are visibly present. For instance in Figure 9(c), the bottom line is not visible anymore, whereas this is the case in Figure 9(a) and (b). This shows the significance of the stitching algorithm as with the algorithm a single position with optimal illumination can be chosen.

  1. Page 10, line 268, "...wavelength..." how did you select those wavelengths? Did you add a filter in front of your HSI camera to select these wavelengths?

No filter is used as the hyperspectral camera captures 25 different wavelengths simultaneously. From this abundance of wavelengths, we then chose three wavelengths that have the largest contrast between them, to illustrate that it is important to carefully select not only the spatial position but also the wavelength to optimize the defect finding.

Page 3, line 93-96. There are a lot of incomplete sentences. 

Line 93-96 have been reformatted to:
This algorithm not only facilitates measurements with heterogeneous illumination sources, which is a common challenge with Hyperspectral Imaging (HSI) or UV cameras, but also provides a solution that is particularly useful in industrial settings. Achieving even illumination can be difficult due to safety or financial constraints in such environments. By applying the same algorithm to reconstruct an image at a specific spatial location, it ensures even illumination, thereby enhancing the detection of defects.

Reviewer 2 Report

Comments and Suggestions for Authors

Ok for this new version

Author Response

Dear reviewer,

Thank you for evaluating the quality of this research paper and finding it up to the standards of this journal.

Best regards,

Thomas